# Next-Gen Stroke Models: The Promise of Assembloids and Organ-on-a-Chip Systems

**DOI:** 10.3390/cells14241986

**Published:** 2025-12-14

**Authors:** Giorgia Lombardozzi, Kornélia Szebényi, Chiara Giorgi, Skender Topi, Michele d’Angelo, Vanessa Castelli, Annamaria Cimini

**Affiliations:** 1Department of Life, Health and Environmental Sciences, University of L’Aquila, 67100 L’Aquila, Italy; giorgia.lombardozzi@guest.univaq.it (G.L.); chiara.giorgi2@graduate.univaq.it (C.G.); vanessa.castelli@univaq.it (V.C.); 2Institute of Molecular Life Sciences, HUN-REN Research Centre for Natural Sciences, 1117 Budapest, Hungary; szebenyi.kornelia@ttk.hu; 3Department of Clinical Disciplines, University of Alexander Xhuvani of Elbasan, 3001 Elbasan, Albania; skender.topi@uniel.edu.al; 4Sbarro Institute for Cancer Research and Molecular Medicine, Temple University, Philadelphia, PA 19122, USA

**Keywords:** stroke models, brain assembloids, organ-on-a-chip, blood–brain barrier, microfluidics, 3D in vitro models

## Abstract

The complexity of stroke pathophysiology, involving intricate neurovascular interactions and dynamic cellular responses, has long challenged the development of effective preclinical models. Traditional 2D cultures and animal models often fail to fully recapitulate human-specific features, limiting translational success. Emerging 3D systems, particularly brain assembloids and organ-on-a-chip platforms, are offering new opportunities to create more physiologically relevant stroke models. Assembloids, which integrate multiple brain-region-specific organoids, enable the study of interregional connectivity and complex cellular responses under ischemic conditions. Organ-on-a-chip platforms, by mimicking key tissue interfaces such as the blood–brain barrier and incorporating controlled fluid dynamics, enable a dynamic and highly customizable microenvironment with real-time monitoring capabilities. This review introduces and characterizes these two cutting-edge platforms (assembloids and organ-on-chip technologies), exploring their potential in stroke research while also discussing current challenges that need to be addressed for their broader adoption in translational applications.

## 1. Introduction

Stroke remains one of the leading causes of death and disability worldwide, with a growing burden particularly in low- and middle-income countries, despite advances in acute therapies and prevention strategies [1,2]. According to the World Health Organization (WHO) clinical definition, stroke was identified and divided into three major pathological subtypes: ischemic stroke, intracerebral haemorrhage, and subarachnoid haemorrhage [1].

Ischemic stroke (IS), the most prevalent subtype, is caused by the occlusion of cerebral blood vessels, leading to reduced blood flow and impaired neuronal function. Its progression is generally divided into three main phases. In the acute phase, lasting from a few hours to days, oxygen and energy deprivation result in the formation of the ischemic core, while apoptotic processes and excitotoxic phenomena dominate in the surrounding penumbra [3]. This is followed by the subacute phase, characterized by increased permeability of the blood–brain barrier and a strong inflammatory response that contributes to lesion expansion. Finally, during the chronic phase, which spans from weeks to months, processes of remodelling and repair of the neurovascular unit (NVU) are activated [4].

Current therapeutic strategies for IS include intravenous thrombolysis and/or mechanical thrombectomy. Thrombolysis is effective within 4.5 h of symptom onset (up to 9 h in selected patients, such as wake-up stroke), while thrombectomy is beneficial in patients with large vessel occlusion up to 6 h, and up to 24 h if selected by advanced imaging. Both treatments are highly time-dependent, and a rapid healthcare system response is crucial to maximize their benefits [2].

In cerebral ischemia, multiple mechanisms are activated, including inflammation, oxidative stress, and excitotoxicity. In particular, neuroinflammation contributes both to the amplification of neuronal damage through the recruitment of immune cells and to the initiation of reparative processes [5]. Although numerous studies have identified potential therapeutic targets for stroke treatment, most of these strategies have failed to demonstrate efficacy in clinical settings. This gap raises concerns about the actual predictive value of experimental models for such a complex disorder. The lack of successful translation from preclinical studies to patients has been attributed to several factors, including species differences, treatment-related side effects, and the high heterogeneity of stroke in humans. This review aims to examine the main experimental models used for studying IS, highlighting their strengths and limitations. The goal is to provide a critical overview of currently available platforms, from traditional in vitro models to advanced systems such as organoids, assembloids, and microfluidic devices. Emphasis is placed on their ability to faithfully recapitulate pathophysiological processes, their utility in drug discovery and preclinical testing, and the challenges that remain in translating findings into clinical applications. Also, recent reviews, including Pang et al., have critically described advances in NVU modelling and the adoption of human cell-based systems for ischemic stroke research [3]. However, these analyses primarily focus on either NVU-centered platforms or the general transition from 2D to 3D in vitro models, without fully addressing the convergence of bioengineering innovations, such as assembloids, organ-on-chip devices, microfluidic vascularization strategies, and 3D bioprinting approaches, that collectively define the next generation of human-relevant stroke models. The present review aims to fill this gap by providing an integrated overview of these emerging technologies and discussing how their combined use may enhance the translational relevance of ischemic stroke modelling.

## 2. When Models Fail: Limitations of 2D and Animal Models in Stroke Research

Despite advances in disease management, stroke remains a major public health and social burden. Current therapeutic options, although effective, are restricted to a limited number of patients due to narrow therapeutic windows and numerous clinical contraindications. As a result, complementary strategies are urgently needed to expand therapeutic opportunities. However, despite decades of research and many promising results in preclinical models, no therapy has yet been successfully translated into clinical practice. This persistent gap between experimental systems and the complex pathophysiology of human stroke underscores the urgent demand for innovative, human-relevant models [6].

Figure 1 summarizes the progressive evolution of experimental models used to study ischemic stroke, from simple 2D systems to advanced human-relevant platforms.

Much of our knowledge of the complex pathophysiology of IS comes from animal models, partly due to the availability of genetically modified strains [7]. Rodents are the most widely used species due to their similarity to humans in both physiology and vascular system [7,8]. In rodents, focal cerebral ischemia models represent a well-established in vivo tool to reproduce key features of human stroke. The most common approach relies on transient or permanent occlusion of the middle cerebral artery (MCAO), using different techniques applied in both rodents and larger mammals [9]. The most common method, first introduced by Koizumi et al. and later refined by Longa et al., involves the insertion of a filament through the external or internal carotid artery up the MCA branch, where blood flow is blocked for a variable time (30–120 min) [10,11]. Reperfusion can then be achieved by simply withdrawing the filament, thereby allowing the modelling of either transient or permanent ischemia [9].

However, the complexity of animal models can sometimes represent a significant limitation when investigating basic mechanisms, which would benefit from the use of simpler systems, such as two-dimensional cell cultures. In vitro models provide the opportunity to reproduce, under controlled conditions, the key features of the ischemic penumbra, an area of tissue that, while retaining residual viability, becomes functionally impaired and thus represents the primary target of therapeutic strategies. To mimic this scenario, several approaches have been developed to recreate ischemia-like conditions and reproduce the cellular alterations observed in vivo [7,8]. The most common method to mimic ischemia-like conditions is oxygen and glucose deprivation (OGD). In this methodology, cell or tissue cultures are placed in a hypoxic or anaerobic chamber where the standard atmosphere is replaced with an N_2_/CO_2_ gas mixture, while they are simultaneously maintained in glucose-free medium, thereby reproducing the combined oxygen and glucose deprivation characteristic of ischemia.

The duration of exposure can range from 30 min to 24 h, depending on the cell type and the level of ischemic damage to be modelled [7,8]. Following the OGD phase, reoxygenation and glucose reintroduction into the medium are performed, allowing simulation of the reperfusion phase, which is known to exacerbate ischemic injury further [12]. While reperfusion is essential to restore the functionality of brain tissue, it can also trigger a secondary injury known as ischemia–reperfusion (I/R) damage [13].

Hypoxia can also be induced through chemical inhibition by targeting the mitochondrial electron transport chain using agents such as antimycin, rotenone, and sodium azide, or via enzymatic induction: a less common approach induces hypoxia by modulating the glucose oxidase–catalase (GOX/CAT) enzymatic system, often in combination with 2-deoxyglucose, thereby reproducing oxygen and glucose restriction [8,13]. Although OGD is a widely used and valuable method for reproducing ischemic injury, it still presents several critical limitations [14]. The first issue concerns oxygen concentration: cell cultures are typically maintained under hyperoxic conditions, around 21% O_2_, compared with physiological values of 2–8% in tissues and 10–13% in arterial blood. This hyperoxic environment can alter cellular sensitivity to oxidative stress and modify the ischemic response [15]. Glucose availability in culture media is another potential source of artifact, as standard formulations often contain more than 20 mM glucose, whereas plasma and brain levels are considerably lower (5.5–7.8 mM and 0.8–2.4 mM, respectively). Chronic hyperglycemia can negatively affect cell viability and interfere with key signalling pathways, such as those mediated by AMPK [16].

For decades, two-dimensional cell cultures have been a fundamental tool in stroke research, providing a simple, low-cost, and highly reproducible experimental system compared with animal models. However, their simplified structure cannot reproduce the complex environment that cells encounter in brain tissue. In vivo, neurons and glial cells are organized into three-dimensional networks, interact with multiple cell types, and establish dynamic relationships with the extracellular matrix. These interactions profoundly influence morphology, survival, differentiation, gene expression, and functional properties such as electrophysiological connectivity. For this reason, three-dimensional brain models are emerging as more physiologically relevant alternatives, capable of more realistically reflecting the complexity of the human brain [17].

An intermediate approach between traditional two-dimensional cell cultures and in vivo models is represented by an organotypic brain slice. These systems preserve much of the tissue architecture and allow different cell types to retain physiological properties, spatial organization, and network connectivity like those observed in vivo [17,18]. However, they also present important limitations: their viability in culture is limited to relatively short periods, and prolonged maintenance leads to progressive degeneration, which restricts their usefulness for long-term studies. In addition, the lack of afferent and efferent inputs leads to synaptic rearrangements that alter the original physiology.

### Bridging the Gap: The Unmet Needs in Stroke Modelling

In recent years, the advent of pluripotent stem cells has enabled the generation of a wide variety of three-dimensional neural models that differ in both complexity and regional specificity. Alongside so-called brain organoids, which can reproduce multiple brain regions and their interactions, more region-specific structures have been developed, including organoids of the forebrain, midbrain, cerebellum, hippocampus, and hypothalamus [19,20]. The generation of these models relies on the modulation of developmental signalling pathways through the addition of patterning factors, which guide cellular differentiation toward distinct regional identities [7]. However, no universal protocol exists; approaches may vary depending on the type of starting cells, the use of extracellular matrix components, and the combination of biochemical cues employed to drive tissue organization. These methodological variations are not trivial, as they directly affect the level of maturation, structural and functional complexity, and ultimately the suitability of each model for specific applications, ranging from disease modelling to drug screening [7,20].

To date, only a limited number of studies have exposed cerebral organoids to hypoxic conditions, and these investigations have primarily focused on how reduced oxygen availability affects early neurodevelopment and cortical formation. The extracellular matrix (ECM) is far more than a passive structural scaffold; it is a dynamic environment that delivers essential biochemical and mechanical cues for cell adhesion, survival, and differentiation. In brain organoid research, recreating an ECM that approximates in vivo conditions is essential for enabling the formation of complex, functionally relevant three-dimensional structures; however, current models still lack several critical features of the native brain ECM [21]. Natural biomaterials such as Matrigel and hydrogels have been widely used to partially mimic these conditions, but their variability and limited compositional control remain significant drawbacks [22].

Another significant challenge concerns the high heterogeneity of brain organoids, which can vary substantially in size, shape, and cellular composition. This variability reduces comparability between experiments and limits the reliability of the results [7,23]. Achieving more uniform and standardized organoids is therefore essential, not only to ensure more controlled experiments but also to make these models suitable for large-scale screening studies and future therapeutic applications.

A significant limitation of brain organoids is their incomplete cellular composition, as they lack several non-neuronal cell types, including endothelial and immune cells. In particular, the absence of microglia is critical, since these cells play key roles in both the immune response of the central nervous system and the maintenance of brain homeostasis. Incorporating these cell types can substantially enhance the ability of these models to recapitulate the complexity of the human brain [7,22].

Another major challenge is the lack of a functional vascular network. Without blood vessels, brain organoids inevitably develop hypoxic and necrotic cores, which limit their growth, maturation, and long-term stability [17,20]. The absence of a vascular system dramatically limits oxygen and nutrient delivery in brain organoids, often resulting in extenxive apoptosis. As a consequence, brain organoids inevitably develop hypoxic and necrotic cores, which limit their growth, maturation, and long-term stability.

Late brain development relies on the establishment of a functional vascular network. In the subventricular zone (SVZ), blood vessels not only provide oxygen and nutrients but also create a regulatory niche for neural progenitors, supporting their survival and differentiation. The lack of adequate vascularization in current organoid models may therefore contribute to the reduced presence of SVZ-like progenitor cells that in the human brain play a central role in neurogenesis and in endogenous repair responses after ischemic injury. This limitation also affects the ability of organoids to recapitulate more complex structures, such as the cortical plate. These observations highlight that the vascular component is not merely a metabolic support but a fundamental driver of proper brain architecture, maturation, and the representation of neurogenic niches relevant to stroke research [20,24].

Brain vascularization is a fundamental developmental event that underpins the brain’s ability to meet its high metabolic needs. This process begins very early in embryogenesis, around the time of neural tube closure (29 to 32 days after fertilization), and continues well into the postnatal stages, gradually establishing an extensive and finely branched network of blood vessels throughout the brain [25,26]. Nutrients and oxygen are supplied to the neuronal tissue through the surrounding amniotic fluid by simple diffusion. Initially, blood vessel growth is driven by angiogenesis originating from external vascular structures, most notably the perineural vascular plexus, which acts as a primary gateway. A coordinated interplay of specialized cells, growth factors, and molecular pathways regulates the process [27]. The vascularization of brain organoids can be achieved through different methods. The most used approach involves co-culturing with endothelial cells (ECs) or their progenitors. The study by Pham et al. confirm the feasibility of vascularizing brain organoids using patient-derived ECs [28]. Patient-derived induced pluripotent stem cell (iPSC)-derived organoids, generated following the Lancaster et al. [19] method, were embedded on day 34 in Matrigel containing ECs differentiated from the same patient’s iPSCs. Interestingly, robust vascularization was observed in the more organized outer layers of the organoid, as evidenced by staining with CD31, a marker of endothelial cells [28]. More recently, Dao et al. demonstrated the development of brain-specific ECs with specialized tight junctions and transporter expression within cerebral organoids after fusing them to blood vessel organoids derived from the same hiPSCs [29]. Shi et al. generated vascularized organoids through their studies by co-culturing them with human umbilical vein endothelial cells (HUVECs). The vascularized organoids were maintained in culture for more than 200 days, resulting in a reduction of the necrotic core that typically characterizes 3D cell models, as demonstrated by the lower number of HIF1-positive cells and caspase-3-positive apoptotic cells. In vascularized organoids, the enhanced growth and improved cell survival observed compared to non-vascularized counterparts help maintain a healthier and more stable tissue architecture before ischemic insult, thereby reducing baseline hypoxia-related damage and improving the reliability of stroke modeling [30]. Another approach involves the use of engineering strategies, such as genome-editing techniques, to provide adequate vascularization to organoids. In the study by Cakir and collaborators, genome editing was applied through the overexpression of the human transcription factor ETS variant 2 (ETV2) in hESCs, successfully reprogramming them into endothelial cells (ECs) and confirming the formation of vascular-like structures within the hCOs [31]. The formation of cerebral blood vessels is tightly regulated by various growth factors that control their development and stability. Incorporating these signals into brain organoid cultures has proven to be an effective way to enhance vascularization. For instance, adding VEGF during early differentiation stages can trigger the formation of vessel-like structures, replicating key features of the blood–brain barrier [32].

## 3. Integrating Bioengineering Tools to Advance Brain Organoid Models

In recent years, advances in bioengineering have opened new perspectives for overcoming the traditional limitations of 3D neural models. The most innovative technologies aim to precisely control the cellular environment, enhancing the maturation, functionality, and reproducibility of brain organoids. These strategies are particularly relevant in the context of modelling complex disorders such as IS, where the ability to capture dynamic processes, including vascular interactions, neuronal network reorganization, and inflammatory responses, remains a major challenge for conventional systems. By integrating microfluidics, 3D bioprinting, biosensors, and organ-on-chip platforms, researchers are moving toward the creation of organoid models that not only mimic the structural complexity of the human brain but also allow real-time monitoring and functional assessment.

### 3.1. Microfluidic Devices

Microfluidic devices represent a versatile platform for recreating complex microenvironments characterized by controlled gradients of oxygen and biochemical signals. In vitro vascular models can be developed using two main approaches: self-organizing (emergent) and pre-patterned (top-down). In the self-organizing approach, three-channel microfluidic devices are often used: a central gel flanked by two perfusion channels. Endothelial cells, with or without stromal cells, are incorporated into the gel or seeded onto the channel walls after gel polymerization. Within a few days, these cells spontaneously self-organize into perfusable vascular networks, mimicking natural vasculogenesis and angiogenesis. The pre-patterned approach, on the other hand, involves creating hydrogel scaffolds using techniques such as 3D printing or sacrificial materials. Predefined channels are then lined with endothelial cells, enabling the formation of functional vascular networks with controlled geometries and orientations, ranging from simple macroscopic conduits to complex microscopic vessels [33]. These devices contain continuously perfused chambers populated with human cells arranged in three dimensions to mimic tissue architecture. Thanks to precise control of the microenvironment, it allows the regulation of extracellular matrix topology as well as nutrient and waste flows [34,35]. Microfluidic platforms constitute the technological foundation upon which more complex physiological constructs, such as organ-on-a-chip systems, are built.

There are different materials and fabrication techniques for developing these models tailored to various diseases, capable of recapitulating the vascular microenvironment (Shakeri et al., 2023 [36]). Polydimethylsiloxane (PDMS) is the most widely used polymer PDMS is distinguished by several advantages, including ease of fabrication, optical transparency, biocompatibility, and gas permeability [36]. One of the main disadvantages of PDMS is its porosity, which makes it prone to absorbing hydrophobic compounds, a factor that must be considered in drug screening studies. Additionally, it is not inert, meaning that silicon can leach into the cell culture environment. For this reason, new approaches have emerged involving the use of alternative polymers, including thermoplastics and polyesters, capable of mimicking the characteristic softness of vascular tissue [35].

A study conducted by Denecke and his group led to the development of a polystyrene-based device to simulate IS conditions in vitro and assess astrocyte activation in the penumbral region [37]. By exploiting the lower gas permeability of polystyrene compared to PDMS, they precisely controlled oxygen and nutrient gradients to recreate the necrotic and penumbral regions typical of stroke. The results showed that both hypoxia and nutrient deprivation contribute to astrocyte death and dysfunction, causing persistent alterations, such as morphological changes and impaired calcium signalling, even after normal conditions were restored [37].

An alternative approach to generate perfusable microvascular networks and study angiogenesis involves culturing endothelial cells in microchannels or chambers filled with natural hydrogels (such as Matrigel, collagen, fibrin, or decellularized ECM). In these systems, endothelial cells are typically placed in a central channel flanked by two parallel channels that supply nutrients and support vessel formation [38].

### 3.2. Organ-on-a-Chip Systems

Organ-on-a-chip (OoC) refers to in vitro microfluidic systems that replicate key features of human physiology and pathology [34].

To achieve more biomimetic vascular structures in organ-on-a-chip models, several approaches have been developed. One strategy involves integrating multiple microfluidic compartments, some of which are filled with biomimetic scaffolds or biological materials such as collagen, to recreate the physiological vascular environment in vitro. Cells can be seeded within these scaffolds and exposed to mechanical stimuli or chemical gradients, inducing the formation of complex vascular networks, including both macro- and microvasculature [39]. These configurations allow for controlled studies of the tissue blood interface, cell–cell interactions, endothelial permeability, and phenomena such as platelet aggregation processes that are difficult to observe in vivo [35]. Various techniques have also been explored to create circular microfluidic channels, which more closely replicate the natural geometry of blood vessels compared to conventional rectangular PDMS channels. These solutions enhance the realism of blood flow and the biological relevance of vascular models [36]. OoC models rely on sophisticated microfabrication techniques that require strict standardization, device-to-device differences, material-dependent effects, and the technical expertise needed to operate microfluidic systems still hinder widespread implementation [40].

Organoids and organ-on-a-chip platforms differ significantly in their approach and level of control. In organ-on-a-chip systems, key physiological functions are recreated within a microfluidic device, where cell types, architecture, and the microenvironment are precisely regulated. Organoids, on the other hand, arise from pluripotent stem cells that self-organize spontaneously into 3D structures. While this self-organization provides a biologically realistic model, it limits the ability to engineer a perfusable vascular network in a controlled manner directly.

### 3.3. 3D Bioprinting

3D bioprinting is a cutting-edge biofabrication technology that combines biocompatible materials and biological components into a single bioink, enabling the three-dimensional printing of structures with customized geometries. This approach allows the creation of complex biological structures, closely replicating the architecture and functions of natural tissues with high fidelity [41]. This approach offers a key benefit: it allows the creation of constructs with mechanical characteristics tailored to the target tissue by depositing bioink layer by layer. At the same time, it provides precise regulation of cell density within the construct, which is essential for accurately reproducing the cellular environment of native tissues for potential biomedical applications.

Recent advances in 3D bioprinting have merged bioengineering precision with the self-organizing potential of stem cell-derived tissues to overcome key limitations of conventional brain organoids. By finely controlling the spatial arrangement of multiple cell types, bioactive molecules, and biomaterials, these techniques can construct highly reproducible and structurally faithful bioengineered organoids. This spatial control ensures not only the external and internal geometry of the constructs but also the alignment and directional arrangement of cells-features that are essential for reproducing neuroarchitecture organization, guiding axonal outgrowth, and enabling communication between different brain regions [42]. A central innovation lies in the fabrication of perfusable vascular networks. High-resolution methods such as two-photon polymerization printing enable the creation of intricate vascular meshes with micropores that facilitate oxygen and nutrient diffusion, reducing hypoxia and apoptosis in the organoid core. Such vascularized models promote dimensional growth and foster long-range neuronal connectivity, supporting advanced cortical organization and the assembly of multi-regional assembloids [43]. One of the central innovations of the study by Xu et al. is the use of two-photon polymerization (TPP) 3D printing to embed a high-resolution artificial vascular network within vascularized organoids, enabling efficient diffusion of nutrients and oxygen to the core. Thanks to this structure, the organoids exhibit volumetric growth, increased cell proliferation, and a significant reduction in hypoxia and apoptosis within their interior. Furthermore, using this printed vascular network, it is possible to construct multi-regional assembloids (e.g., cortical, striatal, and medial ganglionic eminence regions) that show enhanced cell migration, neuronal projections, and excitatory signaling transmitted between different regions [43]. The use of 3D bioprinting for organoid generation is a promising strategy to improve reproducibility and move toward more standardized protocols. Despite notable advances, several challenges remain: the scale gap compared to real organs, the slow printing process, which can impair cell viability due to insufficient oxygen and nutrient supply, and the difficulty of establishing long-term, functional vascular networks [44]. Moreover, while bioprinting offers precise control over cellular arrangement, achieving fully accurate constructs is still complex. Looking ahead, the development of advanced bioinks, optimized biomaterials, and integration with microfluidic culture systems may help overcome these hurdles, paving the way for organoid models that better replicate the complexity and functionality of human organs [44].

### 3.4. Electrochemical Biosensors

Electrochemical biosensors are increasingly being incorporated into advanced in vitro platforms to enhance the functional readouts of ischemia-related processes. When integrated within microfluidic systems, organ-on-a-chip devices, or 3D neural constructs, these sensors provide real-time, label-free detection of biochemical changes that occur during oxygen–glucose deprivation and reperfusion [45].

Unlike conventional endpoint assays, biosensors enable continuous monitoring of parameters such as glucose consumption, lactate accumulation, oxidative stress markers, neuronal damage indicators (e.g., NSE), and fluctuations in oxygen levels. This dynamic profiling is particularly valuable in stroke modeling, where the temporal progression of metabolic failure and recovery is a key determinant of injury severity. The detection of biomarkers offers a less expensive and potentially faster alternative to traditional imaging techniques for stroke diagnosis. Among these, neuron-specific enolase (NSE), an isoenzyme involved in glycolysis in neuronal and neuroendocrine cells, is released into the bloodstream following neuronal damage. Elevated serum NSE levels have been linked to acute cerebral infarction, stroke severity, and infarct volume, as well as to comorbid conditions such as hypertension and possible silent brain injury. However, the clinical use of NSE is limited by conventional detection techniques such as ELISA and ECLIA, which require blood sampling, sample preparation, and trained personnel, factors that hinder real-time diagnosis and monitoring [46]. To address this limitation, Hsu Chen C. and collaborators developed a biosensor capable of accurately and rapidly quantifying serum NSE levels to support the diagnosis and management of acute ischemic stroke (AIS) [46].

Embedding biosensors directly into the culture chamber or perfusion channels allows seamless coupling between physiological events (e.g., BBB disruption, neuronal depolarization, mitochondrial dysfunction) and quantitative electrical signals, improving the sensitivity and temporal resolution of experimental readouts. Such tools are therefore becoming essential components of next-generation NVU-on-chip and perfused organoid models, enabling researchers to track ischemic injury trajectories, evaluate therapeutic responses, and capture subtle functional alterations that are challenging to detect with traditional assays.

### 3.5. Brain Assembloids and Multi-Regional Models

Cerebral assembloids represent a new generation of three-dimensional in vitro models designed to overcome the limitations of traditional organoids. They are created by combining differentiated regional brain organoids or fusing organoids with non-neuronal cell types to investigate interregional interactions within the human brain [47,48]. In other words, whereas a conventional organoid develops from stem cells that self-organize into a single region or tissue, assembloids involve the fusion, integration, or co-culture of two or more distinct regions or cell populations to achieve connectivity, neuronal migration, synaptic interactions, and circuits that more faithfully recapitulate in vivo brain physiology [49]. Based on recent literature, three main categories can be distinguished:Multi-regional assembloids, which combine organoids from different brain areas, such as dorsal cortex and ventral basal ganglia, or thalamus and cortex, to investigate phenomena like interneuron migration and thalamo-cortical circuit formation [50,51]. These assembloids can be generated from healthy human induced pluripotent stem cells (hiPSCs), patient-derived or genetically modified hiPSCs, human embryonic stem cells (hESCs), or even primary tissue [48].Multi-lineage assembloids, which integrate different cell types (e.g., microglia, astrocytes, or vascular cells) to explore neuro-immune or neurovascular interactions, opening new avenues for studying neuroinflammation and the blood–brain barrier [29,52].Functional assembloids, designed to reproduce active synaptic connections and long-range neuronal plasticity, provide advanced platforms for investigating brain connectivity, ischemic injury responses, and preclinical drug testing [51,53].

The multiregional architecture characteristic of brain assembloids, and particularly the presence of interconnected circuits, makes it possible to analyze how neuronal death propagates from one region to another following oxygen–glucose deprivation (OGD). This feature of assembloids clearly distinguishes them from traditional organoid models, where the diffusion of nutrients and oxygen does not accurately reflect normal physiology [53]. This feature also enables the analysis of dynamic processes, such as neuronal migration, circuit reorganization, and neuroinflammatory responses, phenomena that are challenging to reproduce in 2D culture or individual organoids [52]. As demonstrated by the study of Bagley et al., in which they developed a co-culture method combining different brain regions within an organoid tissue, the integration of glial and vascular cells makes it possible to detect neurovascular and neuroimmune interactions, which are important aspects for studying the post-ischemic response [50]. Moreover, the use of assembloids makes it possible to test therapeutic interventions under more physiologically relevant conditions, improving predictive power for preclinical stroke pharmacology [50]. Organoids have demonstrated unique and significant advantages in personalized medicine due to their ability to reproduce key aspects of human diseases. Unlike many other in vitro or in vivo models, these systems can more realistically integrate the combined influence of genetic, metabolic, and environmental factors, which collectively shape the disease’s pathological phenotypes. This capability makes them promising tools for predicting clinical outcomes and supporting the development of more targeted therapeutic strategies [54]. The ability to generate organoids but also assembloids from patient-derived hiPSCs provides the opportunity to create patient-specific models that preserve the donor’s genetic, epigenetic, and potentially phenotypic background. This capability represents a crucial step toward precision medicine in the study of IS. Assembloids, while capable of modeling inter-regional interactions, suffer from high variability in fusion efficiency, cellular composition, and connectivity, which complicates reproducibility across batches. Their increased complexity also limits scalability, making large-scale experiments or high-throughput drug screening challenging [55].

## 4. NVU-on-Chip: An Integrated Model for the Study of IS

Neurovascular-unit-on-chip (NVU-on-chip) systems represent a specialized extension of organ-on-chip technology, leveraging microfluidic platforms to recreate the structural and functional interactions between neurons, glia, endothelial cells, and pericytes. Building upon the principles introduced in the previous sections, these devices apply organ-on-chip engineering specifically to model the NVU, enabling a controlled investigation of ischemia-related mechanisms in a highly integrated and physiologically relevant microenvironment. The NVU represents the fundamental anatomical and functional unit of the brain, composed of cerebral endothelial cells, astrocytes, pericytes, neurons, and the basement membrane. This complex cellular architecture ensures cerebral homeostasis and the proper function of the blood–brain barrier (BBB), playing a crucial role in neuroinflammation, neuroprotection, and repair processes following IS [56,57]. In the context of IS, NVU plays a dual role, acting both as a target of injury and as a key player in recovery processes. NVU disruption leads to structural and functional impairments, including increased blood–brain barrier permeability and the activation of neuroinflammatory responses [57].

The first attempts to model the NVU in vitro were based on two-dimensional cultures of brain endothelial cells, a simple but not very realistic approach. Subsequently, Transwell systems enabled an improvement by introducing co-cultures on semipermeable membranes; however, they remained limited by the lack of flow and reduced direct cell–cell contact [58].

To overcome these limitations, microfluidic platforms were introduced, enabling the growth of stratified 3D cultures under flow conditions. From the early hollow fiber systems, researchers moved to PDMS chips (Booth et al.), characterized by thinner membranes and more physiologically relevant co-cultures. In the following years, several groups adopted similar strategies, increasingly focusing on fully human models, using either primary cells or iPSC-derived cells [59].

Despite these advances, many chips are still impractical, with low throughput. Therefore, there is an unmet need for high-throughput, user-friendly platforms compatible with pharmacological screening.

Vatine et al.’s work presents a human NVU microfluidic model that integrates primary brain endothelial cells, astrocytes, and iPSC-derived neurons, cultured under bidirectional gravitational perfusion [60]. When applied to stroke-like conditions, this system reproduces key pathological features such as loss of blood–brain barrier integrity, reduced mitochondrial membrane potential, and ATP depletion. Thanks to its high throughput and pump-free design, the platform is suitable both for fundamental studies of the NVU and for the evaluation of drug candidates [60].

Recent advances in microfluidics and 3D co-culture technologies have enabled the development of microphysiological systems that better replicate the cellular architecture and microenvironment of human tissues. Among these, the neurovascular unit-on-a-chip (NVU-on-a-chip) has emerged as a particularly promising platform. By recreating the complex interactions between neuronal, glial, and vascular components in a controlled setting, this model addresses many of the translational limitations of animal studies in stroke research [56]. In this model, the aim is to represent the ischemic penumbra, a critical brain region in IS, which, although functionally impaired, retains cellular viability and thus potential for recovery. Within this experimental context, it becomes possible to evaluate the effectiveness of therapeutic approaches, such as cell-based therapies, in promoting post-ischemic recovery. Importantly, NVU-on-a-chip presents a unique opportunity to assess the therapeutic potential of stem cell-based interventions and to generate clinically relevant insights for IS management [56].

These systems can recapitulate the architectural and functional complexity of the human brain, overcoming many of the limitations of traditional in vitro and in vivo models and providing a more realistic perspective on the cellular and molecular dynamics following ischemia. Moreover, the possibility of customizing them using patient-derived hiPSCs opens a previously inaccessible dimension of precision medicine, enabling the study of individual variability and the testing of targeted therapeutic strategies.

### Patient-Derived Models for Precision Medicine in Stroke

The possibility of generating brain models from patient-derived hiPSCs represents a major advancement for precision medicine, and it offers particular value for ischemic stroke by enabling the study of patient-specific susceptibility and therapeutic responses. Technologies such as organoids, multi-region assembloids, and NVU-on-chip platforms, when derived from patient-specific lines, preserve the donor’s genetic and epigenetic background while reproducing in vitro phenotypic variability and individual drug responses. This approach enables targeted testing of neuroprotective compounds for efficacy and toxicity, the identification of predictive biomarkers of outcome, and the exploration of how comorbidities (e.g., hypertension, diabetes) or genetic variants influence ischemic vulnerability.

Recent advances in NVU-on-chip models have shown that post-ischemic recovery relies not only on direct neuronal regeneration but, more importantly, on the restoration of neurovascular unit integrity and functionality [61]. These systems, which integrate endothelial cells, astrocytes, pericytes, microglia, and neurons, recreate in vitro the complexity of the blood–brain barrier and enable a comparative assessment of the efficacy of different stem cell types. Moreover, they provide the opportunity to evaluate regenerative strategies on a patient-specific background, thereby allowing the prediction of individual clinical responses and the development of more targeted therapeutic approaches [61]. As already demonstrated in oncology, patient-derived organoids (PDOs) preserve the donor’s genetic and epigenetic background, faithfully reproducing in vitro the individual variability in treatment responses. This enables the identification of drug sensitivities or resistances and the development of targeted therapeutic strategies, thereby overcoming the limitations of traditional models. The establishment of organoid biobanks from multiple patients could further provide a valuable resource for comparative studies and patient stratification based on predictive biological signatures [44]. The integration of these models with advanced technologies discussed above, such as microfluidic platforms and NVU-on-chip systems, allows the reconstruction of complex physiological microenvironments, including vascular and immune components, which are particularly relevant for studying ischemic pathology (Table 1).

## 5. Discussion and Conclusions

The advancement of in vitro modelling for IS has initiated a profound transformation in neuroscience research, offering increasingly sophisticated platforms to investigate the multifaceted nature of cerebral injury and recovery. Traditional models, including two-dimensional cell cultures and animal systems, have provided foundational insights into stroke pathophysiology, yet they lack the structural complexity, cellular diversity, and dynamic neurovascular interactions that characterize the human brain. The emergence of brain organoids has enabled the study of neurodevelopment and disease in three-dimensional environments, allowing researchers to simulate key aspects of neuronal differentiation, synaptic formation, and cellular stress responses. However, limitations such as the absence of vascular networks, immune components, and interregional connectivity have restricted their translational relevance. To overcome these challenges, the development of multi-regional and multi-lineage assembloids as well as NVU on chip systems has introduced a new level of biological realism, integrating multiple brain regions and diverse cell types to recreate functional circuits and physiologically relevant microenvironments. These models allow for the investigation of neuronal migration, neuroinflammation, blood–brain barrier dynamics, and post ischemic remodelling in ways that were previously unattainable. The incorporation of bioengineering technologies such as microfluidics, three-dimensional bioprinting, and electrochemical biosensors has further enhanced the precision and versatility of these platforms. Microfluidic systems enable the simulation of fluid dynamics, oxygen and nutrient gradients, and controlled perfusion, while bioprinting allows for the spatial organization of cells and the creation of perfusable vascular structures. Biosensors contribute to real time monitoring of molecular and cellular responses, facilitating dynamic assessment of injury progression and therapeutic efficacy. Importantly, the use of patient derived induced pluripotent stem cells introduces a personalized dimension to stroke modelling, preserving the donor’s genetic and epigenetic background and enabling the study of individual susceptibility, drug responsiveness, and the influence of comorbidities such as hypertension, diabetes, and metabolic disorders. Looking ahead, several strategic directions are poised to shape the future of stroke research. These include the standardization of protocols for organoid and assembloid generation to ensure reproducibility and scalability across laboratories, the integration of multi-organ systems to study systemic responses and inter organ communication, the establishment of patient-derived biobanks to support comparative studies and stratified medicine, and the development of regulatory frameworks to facilitate clinical translation. Additionally, the incorporation of functional technologies such as optogenetics, electrophysiological recording, and neuromodulation will enable the interrogation of neural circuits, plasticity, and recovery mechanisms in post-stroke models. As these platforms approach clinical relevance, it is essential to address ethical considerations and societal implications to ensure their responsible and equitable use. In conclusion, the convergence of stem cell biology, tissue engineering, and precision medicine is redefining the landscape of ischemic stroke research. These next generation models offer powerful tools to deepen our understanding of stroke pathophysiology, accelerate the development of targeted therapies, and pave the way for personalized approaches that may ultimately transform clinical care and improve patient outcomes. In recent years, it has become increasingly evident that the development of next-generation in vitro models cannot advance without the integration of advanced analytical tools, particularly artificial intelligence (AI) and multi-omics approaches. Several studies have demonstrated how AI can be used to automate the segmentation, quantification, and morphological tracking of organoids, making 3D phenotypic analysis more objective and scalable while reducing observer-dependent variability [62]. In parallel, the concept of “organoid intelligence” proposes an even closer integration between organoids and AI algorithms, in which stem-cell-derived neural systems and computational methods are co-designed to dynamically investigate human neural networks and extract complex patterns from large volumes of functional and structural data [63]. At the same time, multi-omics approaches are becoming central in the study of cerebrovascular diseases and represent a natural complement to human in vitro models. The application of multi-omics to stem-cell-based therapies for stroke has revealed new signaling pathways involved in neuroprotective and regenerative processes, thereby refining precision-medicine strategies [64]. Looking ahead, the convergence of humanized in vitro platforms, AI-guided analysis, and multi-omics readouts, combined with targeted in vivo validation, represents one of the most promising directions to enhance the ability of stroke models to mimic the biological complexity of the disease and to support the development of personalized therapeutic interventions [65].

This work is a narrative review focusing on conceptual and technological advances in next-generation in vitro models for ischemic stroke. Relevant literature was identified through PubMed and Google Scholar using combinations of keywords including ischemic stroke, organoids, assembloids, neurovascular unit, organ-on-chip, microfluidics, and vascularized organoids. We considered peer-reviewed publications from the past decade and earlier foundational studies when relevant. Given the narrative nature of this review, the aim was not to produce a systematic inventory but to highlight key methodological innovations and influential contributions shaping the development of advanced human-based stroke models.

## Figures and Tables

**Figure 1 cells-14-01986-f001:**
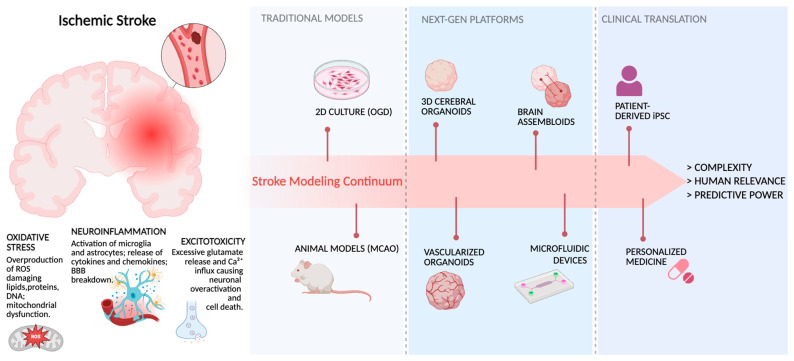
Progressive Evolution of Experimental Models for Ischemic Stroke Research.

**Table 1 cells-14-01986-t001:** Comparative overview of the principal next-generation in vitro models used for ischemic stroke research.

Model Type	Biological Complexity	Key Features	Advantages	Limitations	Stroke-Relevant Applications
**2D cell culture**	★✩✩✩ (low)	Flat monolayers defined and customizable cellular composition, high experimental throughput.	Easy to handle, cost- efficient, reproducible, suitable for mechanistic studies and drug screening.	Lack 3D architecture, limited cell-cell and cell-matrix interactions, low physiological relevance.	OGD/OGD-R assays, neuroprotection screening, oxidative stress and apoptosis pathways.
**Brain organoids**	★★✩✩ (medium)	3D self-organized structures, early neurodevelopmental features, regional specification possible.	Human-specific architecture, recapitulate neurogenesis and cortical layer formation, long-term culture.	High variability, limited vascularization, restricted maturation, weak NVU representation.	Modeling developmental susceptibility to ischemia, cell-type specific responses, personalized hiPSC models.
**Assembloids**	★★★✩ (medium high)	Fusion of region-specific organoids, long-range connectivity.	Mimic inter-regional communication, better network-level physiology.	Variability in fusion and connectivity, limited scalability, still non- vascularized	Studying propagation of ischemic stress between brain regions, circuit-level vulnerability.
**Microfluidics-devices**	★★✩✩ (medium)	Microscale channels, controlled flow, gradients and microenvironment.	High precision, recreate shear stress, nutrient flow, O_2_/glucose gradients.	Require technical expertise, limited multicellular complexity.	Modeling perfusion deficits, OGD-R kinetics, real-time barrier assessment.
**3D Bioprinting**	★★✩✩ (medium)	Leyered, spatially controlled printing of cells + ECM biomaterials.	Tunable architecture, controlled cell orientation, reproducible geometry.	Complex protocols, limited maturation, ECM bioinks not fully brain-like.	Testing neuroprotective scaffolds, oxygen diffusion patterns, cell-specific survival.
**Electrochemical biosensor**	★★★✩ (medium-high)	Real-time monitoring of metabolic and injury markers directly within model	High temporaly resolution non-desctructive measurements, integrates with OoC systems.	Tipically require custom engineering, may need calibration.	Tracking lactate, glucose, ROS, barrier integrity during OGD/R.
**Nvu-on-a-chip**	★★★★ (high)	Spatially organized ECs, pericytes, astrocytes ± neurons; perfusable barrier.	Recreates BBB physiology, quantitative permeability readouts, dynamic monitoring.	Still simplified vs in vivo NVU, material-related constraints.	BBB breakdown, leukocyte trafficking, vascular inflammation, reperfusion injury.

## Data Availability

No new data were created or analyzed in this study.

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
