# Peer review of "Next-Gen Stroke Models: The Promise of Assembloids and Organ-on-a-Chip Systems"

_cells, 2025, doi:10.3390/cells14241986_

Round 1
Reviewer 1 Report
Comments and Suggestions for Authors
The authors provide an overall well written review on "Next-Gen Stroke Models". Some reflection on recent reviews on the topic (like the not mentioned review by Pang Pang B, Wu L, Peng Y. on In vitro modelling of the neurovascular unit for ischemic stroke research: Emphasis on human cell applications and 3D model design. Exp Neurol. 2024 Nov;381:114942. doi: 10.1016/j.expneurol.2024.114942. and similar reviews) would have been appreciated in the introduction. What is missing in these reviews (knowledge gap) that provides a rational for this review by the authors?
Additional minor suggestions for improvements are:
L102-104: "In this methodology, cell or tissue cultures are placed in a hypoxic or anaerobic chamber, where the standard atmosphere is replaced with an N₂/CO₂ mixture lacking glucose" Comment: it seems two parameters are changed: the gas composition and the medium composition (lower glucose). However the sentence seems to suggest that via N2/CO2 composition also glucose lacks in the system, which is not clear. Please check and revise for clarity.
L113-115 : "a less common method that relies on manipulating the glucose oxidase and catalase system (GOX/CAT) and 2-deoxiglucose [8], [13]." Please revise the sentence as the grammar makes the sentence difficult to follow. Are there three factors manipulated or two (i.e. "glucose oxidase and catalase system equals one factor")? Please clarify.
L 140 : "short periods, maturation remains incomplete, and the availability of human tissue is very..." Comment: this part refers to brain slices. Slices are used generally thanks to their matured states. Please clarify.
L159-162 :"..modelling to drug screening [7], [20]. To date, only a few studies have exposed cerebral organoids to hypoxic conditions. Most of this research has focused on investigating the effects of reduced oxygen availability on neurological development and cortical formation." Comment: It is unclear what "most of this research" refers to, does it refer to the few studies on low oxygen-exposed cerebral organoids ? Please revise.
L165-167 : "In brain organoid research, accurately recreating the ECM is critical for enabling the formation of complex, functionally relevant three dimensional structures [21]. " Comment: These lines pinpoint on a critical aspect, the ECM which appears in many papers to be modeled "accurately" via brain organoids (however it is a requirement for accuracy of the model to get the ECM right), as authors also state in the following lines (L170-173, see below this comment) that actually a limitation in building ECM is concerned with size, shape and cellular composition. Hence, it needs to be revised in the text that "Brain Organoids" do not YET accurately represent ECM. Careful revision of lines 165-167 with the information that most of these brain Organoids models aim for a "more in-vivo like representation of the tissue" but still like important aspects of the ECM could make readers more alert.
L170-173 : "Another significant challenge concerns the high heterogeneity of brain organoids, which can vary substantially in size, shape, and cellular composition. This variability reduces comparability between experiments and limits the reliability of the results [7], [23]."
L180-181: "Incorporating these cell types is therefore essential to create models that more accurately reflect the complexity of the human brain [7], [22]. " Comment: maybe using the word "essential" is a bit to strict. There could be other means to reflect the complexity of the human brain...., though it is semantics.
L192-195: "The lack of adequate vascularization in current organoid models may
therefore account for the limited abundance of SVZ progenitors and the challenges in recapitulating complex structures such as the cortical plate. This highlights that the vascular component is not merely a metabolic support, but a fundamental driver of proper brain architecture and maturation [20], [24]." Comment; for this part it is not clear why next gen stroke models need SVZ progenitors. please provide context.
L221-223: "Additionally, greater growth was observed in the vascularized organoids compared to the non-vascularized ones, confirming that vascularization promotes embryonic stem cell proliferation and neuronal survival [30]." Comment: While the above lines L221-223 are interesting, it is difficult to appreciate why this information (promoting embryonic stem cell proliferation) is relevant to stroke modeling. It would be helpful to provide context for this statement closing this information gap between the statement made in L221-223 and the reviews objective in reviewing stroke modeling.
L 236: "...overcoming the traditional limitations of 3D neural models. The most innovative" Note: There is an underscore in the next between "the" and "traditional" appearing in the pdf. Please remove.
L 273-289 : "3.2. Organ-on-a-Chip Systems
Organ-on-a-chip (OoC) refers to in vitro microfluidic systems that replicate key
features of human physiology and pathology [36]. These devices contain continuously perfused chambers populated with human cells arranged in three dimensions to mimic tissue architecture. Thanks to precise control of the microenvironment, OoC allow the regulation of extracellular matrix topology as well as nutrient and waste flows [36], [37].
There are different materials and fabrication techniques for developing OoC models tailored to various diseases, capable of recapitulating the vascular microenvironment (Shakeri et al. 2023). Polydimethylsiloxane (PDMS) is the most widely used polymer for creating these models. PDMS is distinguished by several advantages, including ease of fabrication, optical transparency, biocompatibility, and gas permeability [38]. One of the main disadvantages of PDMS is its porosity, which makes it prone to absorbing hydrophobic compounds, a factor that must be considered in drug screening studies. Additionally, it is not inert, meaning that silicon can leach into the cell culture environment. For this reason, new approaches have emerged involving the use of alternative polymers, including thermoplastics and polyesters, capable of mimicking the characteristic softness of vascular tissue [37]. " Comment: The L73-288 refer to technical capabilities that are equally true for Microfluidic Devices, discussed in section 3.1 (L 246 and following). In this sense as described in L273-288 there is no distinction between Microfluidic devices and Organ-on-Chip; in fact Organ-on-Chip are microfluidic devices used for organ-tissue mimicry and are based on the capability of designing & fabricating Microfluidic device. Please revise for clarity these sections. Maybe this could be done by simply moving these lines of text up to section 3.1 with providing a bit more context that Microfluidic devices are subsequently providing precursors for Organ-on-Chip technology (which actually authors also mention in Line 307).
L 298-301 : ". Some approaches have also employed ex vivo platforms, culturing isolated arteries under physiological conditions to analyse, in real time, structural and functional changes in the vascular wall[37]." Comment: besides the space missing between wall and [37], these lines describe a completely different means of modeling than Organ-on-Chip, despite that ex vivo platforms can merge potentially with Organ-on-Chip technology and add additional complexity this fact is not made clear by the stated text. Please revise or skip introducing ex vivo platforms in section 3.2.
L326-329: "This spatial control ensures not only the external and internal geometry of the constructs but also their cellular orientation, which is critical for mimicking in vivo neuroarchitecture and maintaining communication between distinct brain regions [41]." Comment: In the Lines 326-329 it is unclear what is meant with cellular orientation. The definition of "cell orientation" has not been introduced before and it is also not further explained after these lines. Please provide appropriate context.
L 354: "3.4. Electrochemical biosensors " Comment: This section is making a general statement on "diagnostics" during stroke diagnosis/therapy but does not link to the objective of the review on "stroke models. Please revise accordingly.
L 418: "4. NVU-on-chip: an integrated model for the study of IS" Comment: In this line a new type of advanced model for IS is introduced. This section should be opened with a few lines of providing context to the previous subsection 3.2 on Organ-on-Chip technology used in this field of brain models; since the NVU-on-chip is simply a specific application of what is doped Organ-on-Chip; authors should make clear in this section 4, that readers can appreciate this and that the NVU is also originating from exploring microfluidics for modeling purposes of the NVU.
L469-470: "The possibility of generating brain models from patient-derived hiPSCs represents a breakthrough for precision medicine in ischemic stroke. Technologies such as organoids..." Comment: this is not only a break through for IS but in general for precision medicine, which then includes IS. Please revise.
L 476: "hypertension, diabetes) or genetic variants influence ischemic vulnerability." Remove the underscore at the end of the sentence as appearing in the pdf in this line.
Reviewer 2 Report
Comments and Suggestions for Authors
The topic of the review is very interesting and relevant. I found the review informative and useful.
The review describes the current state of the problem and prospects for further research.
However, I cannot recommend the manuscript for publication, as it is not without its shortcomings.
Main comments
1.
I have been following this topic for a long time and I have my own selection of articles. The number of published works is enormous. I'm not just interested in stroke models. Perhaps for this reason, in my collection and in the reference list of the manuscript under consideration, I found only one identical publication.
I attempted to find additional articles missed in my collection by searching Google and Scholar Google using keywords from the peer-reviewed manuscript. I looked through over a hundred papers and again found only one identical publication.
The authors reviewed a total of 58 publications. What were the criteria for this choice?
I must quote one paragraph from the Instructions for Authors:
“Review: Review articles provide concise and precise updates on the recent progress in a given area of research. Systematic reviews should follow PRISMA guidelines.”
I admit I don't like the PRISMA guidelines, but I encourage authors to follow them.
In any case, authors should indicate the criteria for selecting articles.
2.
The same section of the Instruction for Authors states:
“The main text of review papers should include at least two figures or tables.”
The authors provide only one figure. I believe they should have illustrated a wide range of different models and technologies.
Minor comments
3.
The authors need to delete lines 540–542 and 545–556.
4.
At the end of the abstract, the authors wrote: “This review introduces and characterizes these two cutting-edge platforms, …”
By this point, it's hard to remember what "these two platforms" mean.
Number "these two platforms" earlier and somehow emphasize it.
Reviewer 3 Report
Comments and Suggestions for Authors
This review provides a timely and comprehensive overview of the emerging 3D in vitro models - specifically brain assembloids and organ-on-a-chip systems - in the context of ischemic stroke research. The authors effectively highlight the limitations of traditional 2D and animal models and convincingly argue for the adoption of more physiologically relevant human-based platforms. The manuscript is well-structured, clearly written, and supported by a substantial body of recent literature. The integration of bioengineering tools such as microfluidics, 3D bioprinting, and biosensors is particularly well-articulated, and the emphasis on patient-derived models for precision medicine adds significant value. Below are some minor revisions the authors should respond to before it can be accepted for publication.
- Clarify the distinction between organoids and assembloids early in the text. While the latter part of the manuscript does a good job explaining assembloids, a clearer definition or comparative table early on (e.g., in the Introduction or Section 2) would help readers unfamiliar with the terminology.
- Please expand on the limitations of assembloids and OoC models. The review rightly emphasizes the advantages of these systems, but a more balanced discussion of their current limitations - such as scalability, reproducibility, and the challenge of fully replicating in vivo complexity - would strengthen the critical perspective.
- The authors are suggested to consider adding a summary table comparing the key features, advantages, and limitations of 2D cultures, animal models, organoids, assembloids, and OoC systems. This would enhance the pedagogical value of the review.
- The authors are suggested to strengthen the conclusion by briefly mentioning future directions beyond those already listed - e.g., integration of AI, multi-omics approaches, or in vivo validation of in vitro findings.
- As a review paper, there is only one figure in the maintext, the authors are strongly suggested to add more figures to this paper, for example, pictures of microfluidic chips or 3D printed models for simulating stroke.
Round 2
Reviewer 2 Report
Comments and Suggestions for Authors
As I wrote in my previous report, I found the review informative and useful.
The authors fully complied with all my requirements.
Now I can recommend the manuscript for publication.
Reviewer 3 Report
Comments and Suggestions for Authors
Thanks for the modifications.